# Validating EEG, MEG and Combined MEG and EEG Beamforming for an Estimation of the Epileptogenic Zone in Focal Cortical Dysplasia

**DOI:** 10.3390/brainsci12010114

**Published:** 2022-01-14

**Authors:** Frank Neugebauer, Marios Antonakakis, Kanjana Unnwongse, Yaroslav Parpaley, Jörg Wellmer, Stefan Rampp, Carsten H. Wolters

**Affiliations:** 1Institute for Biomagnetism and Biosignalanalysis, University of Münster, 48149 Münster, Germany; antonakakismar@gmail.com (M.A.); carsten.wolters@uni-muenster.de (C.H.W.); 2Computing Sciences, Faculty of Information Technology and Communication Sciences, Tampere University, 33014 Tampere, Finland; 3School of Electrical and Computer Engineering, Technical University of Crete, 73100 Chania, Greece; 4Ruhr-Epileptology, Department of Neurology, University Hospital Knappschaftskrankenhaus, Ruhr-University, 44892 Bochum, Germany; Kanjana.Unnwongse@kk-bochum.de (K.U.); Joerg.Wellmer@kk-bochum.de (J.W.); 5Department of Neurosurgery, University Hospital Knappschaftskrankenhaus, Ruhr-University, 44892 Bochum, Germany; yaroslav.parpaley@kk-bochum.de; 6Department of Neurosurgery, University Hospital Erlangen, 91054 Erlangen, Germany; stefan.rampp@gmail.com; 7Department of Neurosurgery, University Hospital Halle (Saale), 06097 Halle, Germany; 8Otto Creutzfeldt Center for Cognitive and Behavioral Neuroscience, University of Münster, 48149 Münster, Germany

**Keywords:** epilepsy, beamformer, MEG, EEG, regularization, validation, source analysis

## Abstract

MEG and EEG source analysis is frequently used for the presurgical evaluation of pharmacoresistant epilepsy patients. The source localization of the epileptogenic zone depends, among other aspects, on the selected inverse and forward approaches and their respective parameter choices. In this validation study, we compare the standard dipole scanning method with two beamformer approaches for the inverse problem, and we investigate the influence of the covariance estimation method and the strength of regularization on the localization performance for EEG, MEG, and combined EEG and MEG. For forward modelling, we investigate the difference between calibrated six-compartment and standard three-compartment head modelling. In a retrospective study, two patients with focal epilepsy due to focal cortical dysplasia type IIb and seizure freedom following lesionectomy or radiofrequency-guided thermocoagulation (RFTC) used the distance of the localization of interictal epileptic spikes to the resection cavity resp. RFTC lesion as reference for good localization. We found that beamformer localization can be sensitive to the choice of the regularization parameter, which has to be individually optimized. Estimation of the covariance matrix with averaged spike data yielded more robust results across the modalities. MEG was the dominant modality and provided a good localization in one case, while it was EEG for the other. When combining the modalities, the good results of the dominant modality were mostly not spoiled by the weaker modality. For appropriate regularization parameter choices, the beamformer localized better than the standard dipole scan. Compared to the importance of an appropriate regularization, the sensitivity of the localization to the head modelling was smaller, due to similar skull conductivity modelling and the fixed source space without orientation constraint.

## 1. Introduction

In about 30% of patients suffering from focal epilepsy, pharmacotherapy with anti-epileptic drugs is insufficiently effective [1]. Persisting seizures, anti-epileptic drug side effects, as well as psychiatric comorbidities considerably impact quality of life [2]. While testing other drugs is often inefficient, surgery has a high seizure-freedom rate after one year [3,4]. However, among other issues, seizure recurrence within 2–5 years after surgery in more than 40% of patients is a major problem [5]. A possible explanation is that the brain tissue able to generate seizures (often referred to as epileptogenic zone) has been resected, destroyed, or disconnected only incompletely or that it has been (more or less narrowly) missed. As an advanced diagnostic tool, magnetoencephalography (MEG) and/or electroencephalography (EEG) source analysis may contribute to resolve this issue with their high temporal and good spatial resolution. In particular, the relative insensitivity of the MEG localization to the variability of skull conductivity [6] makes it a promising modality, allowing early identification of surgery candidates [7]. It improves planning and results of invasive recordings [8,9], yields non-redundant information in up to about 30% of cases, and is confirmatory in an additional 50% [10,11]. While EEG measures the same underlying activity, it gives complementary information to the MEG [12,13,14]. Thus, analyzing both MEG and EEG together may give a more complete picture than the single modalities.

Source imaging in both modalities needs a model of the head and accurate numerical EEG and MEG forward modelling methods to compute an estimate of the brain activity [15]. Using magnetic resonance imaging (MRI), an individual head model can be constructed. Segmenting the head into different tissues and using appropriate conductivity values for these tissues will give more accurate localization results, the closer the modelling is to reality [15]. In this context, for example, it has been investigated in which situations distinguishing skull compacta and spongiosa instead of modelling a homogenized skull compartment is important [16,17]. The role of cerebrospinal fluid (CSF), gray matter, and white matter has also been discussed [17,18,19,20,21]. In practice, however, the conductivity values of the tissues are only an estimate and vary widely [22] and the white matter compartment has inhomogeneous and anisotropic conductivity [17,23,24]. Uncertainty analysis showed that especially skull conductivity is a sensitive parameter for EEG, while it does not influence the MEG much [6,25,26]. Skull conductivity has furthermore been shown to be an individually-varying, age-related parameter that can be estimated individually in a skull conductivity calibration procedure from combined somatosensory evoked potential (SEP) and field (SEF) data [6,27]. Here, we will investigate the influence of such volume conduction effects and of skull conductivity calibration on the localization of the epileptogenic zone in two patients with focal epilepsy with postoperative seizure freedom, where the distance of the localization of interictal epileptic spikes to the resection cavity resp. radiofrequency thermocoagulation lesion is used as a reference for good localization. Therefore, we will use a finite element method (FEM) based EEG and MEG forward modelling [28,29] because of its ability to model the above discussed tissue conductivity inhomogeneity and anisotropy.

Due to the non-uniqueness of the EEG and MEG inverse problem, it is, however, well-known that the chosen prior information about the underlying source activity is in most cases an even more sensitive parameter [15]. The quasi standard to solve the source analysis inverse problem in the field of epilepsy is the single dipole reconstruction, either by dipole fit or dipole scan, which might perform well in case of dipolar EEG or MEG topographies [11]. Both dipole fit and scan have the goal to determine the source for which the residual variance (RV) between the measured and the forward-simulated data are minimal. Here, we prefer the dipole scan, also called deviation scan [30], which fully scans a given source space with a fixed number of discrete source positions in the gray matter compartment. It thereby reliably determines the global minimum of the RV cost function over the given source space, also in combination with a head model that distinguishes CSF, gray matter, and white matter. Dipole scanning thus seems more robust than dipole fitting, where the optimization might get stuck in local minima of the RV cost function and which is most often used in combination with a homogenized brain compartment. Further scanning approaches to the inverse problem are beamformers. Beamformer approaches solve the inverse problem of MEG and EEG by using spatial filters and do not need the number of underlying sources as prior information [31]. They are widely used in brain signal analysis, both in the frequency [32] and temporal domain implementation [33]. In epilepsy, they have been used to detect and localize spikes [34,35], to analyze high frequency oscillations [36,37], and for connectivity analysis [38,39]. One parameter in the beamformer analysis is the regularization strength [32,40], which widens the filter profile to avoid missing sources due to source space errors at the cost of less noise suppression. The optimal value depends on the number of sensors, the noise, and the signal of interest, which makes it a difficult problem in practice [40]. The estimation of the data covariance matrix is another important factor that determines beamformer performance. For arbitrary data, the same data are used to calculate the matrix and the reconstruction [31]. In our case, the averaged epileptic spikes were used for this purpose. For event-related data, Refs. [41,42] suggest to use trial data to improve the estimation. The latter has been called the event-related beamformer, while the former is called the average-based beamformer to emphasize the signal type in this work.

In this work, we demonstrate the practical effects and problems of head model selection and the choice of the inverse problem solution on the MEG, EEG, and combined MEG/EEG localization in case studies with two patients with focal refractory epilepsy. We test beamformer usability and robustness, as well as the best regularization parameter choice in comparison to a dipole scan. The distance of source localizations of interictal epileptic spikes to the resection volumes are used as a reference for good localization. This was chosen due to its clinical significance, albeit limitations due to the size of the resection volume, the interictal nature of the evaluated activity [43] and differences between distribution of epileptogenic neurons and MRI correlate of FCDs [44,45]. In the following, the patients and the methods used are presented in Section 2. We will then show our results in Section 3, discuss them in Section 4, and conclude our study in Section 5.

## 2. Patients and Methods

### 2.1. Patients

Patient 1 was, at the time of the measurements, a 29-year-old female suffering from pharmaco-resistant epilepsy. Her semiology was a somatosensory aura of the left arm followed by tonic-clonic movements of the left arm and hand. EEG and MEG data were evaluated by a board-certified epileptologist, who marked 248 interictal spikes. MRI measurements revealed a bottom of sulcus FCD IIb of 1.2 cm^3^ lesional volume according to high resolution 3D-FLAIR and ZOOMit in the right superior parietal lobule. It was confirmed by MEG and EEG source analysis of the marked spikes as a potential localization of the epileptogenic zone. The lesion and surrounding tissue (about 8 cm^3^) was surgically removed in February 2018. The patient was seizure free for one year and did not have any further follow-up.

Patient 2 was, at the time of the measurements, a 17-year-old male presenting with refractory seizures out of sleep, consisting of vocalization followed by right head version, asymmetrical tonic stiffening with right arm extension with right extension and generalized tonic-clonic seizures. After guiding from PET and MEG and EEG source analysis, a very small (about 0.4 cm^3^) FCD type IIb lesion was suspected in the anterior part of the left superior frontal sulcus on MRI. Intracranial stereo-EEG recordings with six depth electrodes implanted in and around the MRI abnormality revealed interictal activity typical for an FCD with seizure onset from the abnormality. The patient was subsequently treated with stereotactically guided radiofrequency thermoablation in August 2017 [46], removing about 1.4 cm^3^. The patient was seizure free for one year and did not have any further follow-up.

### 2.2. Data Acquisition

#### 2.2.1. System Setup

We used a standard EEG system with 74 channels and common average reference. Due to noise, only 66 channels could be used for patient 1 while 70 channels were used for patient 2. Prior to the measurements, EEG electrode positions were digitized using a Polhemus digitizer (FASTRAK, Polhemus Incorporated, Colchester, VT). The MEG system (CTF Omega 2005 MEG by CTF, https://www.ctf.com/, last accessed on 13 January 2022) had 275 gradiometers, 4 of which defective, and 29 reference sensors. No working channels had to be excluded. The MEG reference coils were used to calculate first-order synthetic gradiometers to reduce the interference of magnetic fields originating from distant locations. During the acquisition, the head position inside the MEG was tracked via three head localization coils placed on the nasion, left and right distal outer ear canal. To get the best concordance to the MRI, which was performed in supine position, the patients were also measured in the supine position inside the MEG to reduce head movements and to prevent CSF effects due to a brain shift when combining EEG/MEG and MRI [19]. The data were measured with a sampling rate of 2400 Hz.

#### 2.2.2. Measurement Protocol

For individual skull conductivity estimation and head model calibration [6,47], the patients first underwent a somatosensory evoked potential (SEP) and field (SEF) EEG and MEG recording with electric stimulation of the median nerve. For this purpose, two electrodes were positioned over the right wrist of the patients until thumb movements were visible. The stimulus strengths were then determined to be just above the motor threshold. The stimulus was a monophasic square-wave electrical pulse with a duration of 0.5 ms. We used a random stimulus onset asynchrony between 350 and 450 ms to avoid habituation and to obtain a clear prestimulus interval. The duration of the somatosensory experiment was 10 min for a measurement of 1200 trials and data were acquired with a sampling rate of 1200 Hz and online low pass filtered at 300 Hz.

For recording of interictal epileptic spikes, 5 runs of combined EEG and MEG of 8 min each of eyes-closed resting state were measured, in which the patients were advised to relax. Finally, a board-certified epileptologist inspected these measured EEG and MEG data and marked epileptiform spikes.

### 2.3. Data Preprocessing

#### 2.3.1. MRI Segmentation and Source Space

To compute the forward solution, the head has to be modeled as a volume conductor. While standardized head models exist, creating an individual model will give more precision, which is often needed in the presurgical evaluation of epilepsy. We use three different MRI protocols as a base to model a typical 3 compartment model (3C: skin, skull, brain) and a more realistic 6 compartment model (6C: skin, skull compacta [SC], skull spongiosa [SS], CSF, gray matter [GM], and white matter [WM]). The T1 weighted (T1w) MRI is used to distinguish skin, gray matter, and white matter, while T2 weighted (T2w) MRI allows a better segmentation of the CSF and skull. Additionally, diffusion tensor imaging (DTI) is used to model white matter conductivity anisotropy [48,49]. The complete segmentation procedure is explained below.

The T1w MRI was segmented into a *skin1* mask and a general *brain1* mask, which was additionally segmented into *GM1*, *WM1*, and *CSF1*. This step was performed using SPM12 (https://www.fil.ion.ucl.ac.uk/spm/software/spm12/, last accessed on 13 January 2022) via the FieldTrip toolbox (http://fieldtriptoolbox.org, last accessed on 13 January 2022) [50]. Then, the T2w MRI was registered to the T1w MRI using a rigid registration approach and mutual information as a cost-function with FSL (https://fsl.fmrib.ox.ac.uk/fsl/fslwiki/, last accessed on 13 January 2022). The registered T2w image was then segmented into *skull2*, *CSF2*, and *brain2*. *Skull2* was further separated into skull compacta, *SC2*, and skull spongiosa, *SS2*, based on Otsu thresholding [51].

Subsequently, the tissue masks were combined to create a realistic model. The first step was to prevent any overlap between *GM1* and *WM1* with *SC2*. Then, *CSF* was finalized by combining *brain2* and *CSF2* with the *GM1* and *WM1*. Unrealistic holes were detected and filled by using the MATLAB imfill function. Region detection and thresholding (bwconncomp and regionprops) was then used to ensure a realistic segmentation. The final result was visually inspected to detect and prevent computational errors. Following the recommendations of [52], we cut the model along an axial plane below the skull to avoid an unnecessary amount of computational work.

To model white matter anisotropy, DTI data were processed to reduce eddy current and nonlinear susceptibility artifacts [53,54] with FSL and the SPM12 subroutine HySCO (http://www.diffusiontools.com/documentation/hysco.html, last accessed on 13 January 2022). Diffusion tensors were then calculated and transformed into conductivity tensors by the effective medium approach [48,49].

In the last step of the conductor modelling, a 1 mm resolution mesh was constructed using SimBio-VGRID (http://vgrid.simbio.de/, last accessed on 13 January 2022) [55].

Next, the source space points were placed at the center of the gray matter compartment with approximately 2 mm resolution without orientation constraint. Each source needed to fulfill the Venant condition, i.e., for each source node, the closest FE node should only belong to elements labeled as GM. The fulfillment of this condition is important to prevent numerical errors and unrealistic source modelling for the chosen Venant dipole modelling approach [17,55,56]. The source space was the same for both 3 and 6 compartment models. After setting the tissue conductivities, see Section 2.3.2 and Section 2.3.3, we used SimBio (https://www.mrt.uni-jena.de/simbio/index.php/Main_Page, last accessed on 13 January 2022) and DUNEuro (http://duneuro.anms.de/, last accessed on 13 January 2022) [57] to calculate the 3 and 6 compartment models for patients 1 and 2.

#### 2.3.2. Tissue Conductivity

In the three compartment (3C) head models, we used standard conductivity values of 0.43 S/m [18,58], 0.01 S/m and 0.33 S/m [30,59] for the homogenized skin, skull and brain compartments, respectively. The chosen skull conductivity was found as an optimal choice (average over four subjects) to approximate the skull’s layeredness in compacta and spongiosa [58].

For the 6 compartment (6C) models, we also used 0.43 S/m for skin [18], calibrated individually for skull conductivity as described in Section 2.3.3, chose 1.79 S/m for CSF [60] and 0.33 S/m for GM [18] and the white matter conductivity was modeled using the DTI as described in Section 2.3.1.

#### 2.3.3. Calibration

Following [6,27,54], the skull conductivity was not predetermined by literature values, but estimated individually for each patient from the measured SEP and SEF data in order to, in a second step, facilitate a combined EEG/MEG analysis of the interictal spikes. This type of experiment was chosen because the origin of the somatosensory P20 component is known to be a focal, lateral, and mainly tangentially oriented source and can therefore be well localized by MEG [61,62,63]. Since in contrast to the EEG [64,65,66,67,68] the MEG is nearly unaffected by skull conductivity [6,15,25], the MEG localization can serve as a ground truth to determine the skull conductivity parameter that gives the best fit to the data.

In short, the calibration procedure is as follows [6,27,54]: A dipole scan is used to reconstruct the source underlying the peak of the 20 ms post-stimulation SEF component. Then, after fixing the MEG location, the EEG is used to determine the dipole orientation. Lastly, with the fixed location and orientation, the dipole amplitude is determined from the MEG and the Residual Variance (RV) of the SEP P20 component is stored. Repeating this for different skull conductivities, the conductivity with the lowest RV, leading to the best fitting dipole for both SEP and SEF 20 ms poststimulation components, is then defined as the individually calibrated skull conductivity. To avoid overfitting, we only calibrated for skull compacta (SC) conductivity, while keeping the ratio to skull spongiosa (SS) conductivity, i.e., SC:SS, fixed to 1:3.6 [69].

#### 2.3.4. Data Preprocessing

We processed the epileptic spikes EEG and MEG data using FieldTrip [50]. The raw data were filtered (two-pass zero-phase Butterworth IIRC filter of sixth order) with a high-pass of 2 Hz, a low-pass of 80 Hz, and a notch filter of 50 Hz. Trials were created with 0.5 s of data before and after the peak of the spikes. These trials were checked for similar topographies and then averaged to improve the signal-to-noise ratio (SNR) of the interictal spike patterns. We did not process the MEG and EEG data any further, as spikes were already only marked in visually artifact-free parts of the data. To combine MEG and EEG, we followed [6] regarding the head model calibration and [30] regarding the following normalization procedure: The noise level was estimated from the averaged spike data from −500 ms to −300 ms, relative to the spike peak, which was used centered at 0 ms. Each sensor was normalized by its own noise standard deviation, giving a unit-free measurement. The related leadfields were normalized by the same factor. Then, the combined data were used in the same manner as the single modality data.

#### 2.3.5. Averaging and Time Points of Interest

To localize the averaged spike, a time point has to be chosen. As the activity is not necessarily constant in its location from the start to the peak of the spike, known as propagation [70,71], earlier time points might offer a better estimation of the epileptogenic zone. However, the signal strength at the spike onset is low and the localization thus less reliable. To balance the possible propagation of the spike with the need for signal strength, the middle of the rising flank has been recommended as the time point for localization [70,71].

### 2.4. Inverse Problem and Comparison

#### 2.4.1. Beamformer Filter Design

The idea of a beamformer filter [31] *W* is to suppress every type of signal except for one matching a given forward solution. As this is mathematically impossible for arbitrary types of noise signals, beamformers adapt to the data to suppress only those sources of noise that have been active during the measurement. Therefore, for each location in the source space, a filter *W* with orientation 
ϕ
 is constructed as

(1)
$$W_{\phi }={\mathrm{argmax}}_{\phi }\;{\mathrm{argmin}}_{W}W^{T}CW\;\mathrm{subject}\;\mathrm{to}\;W^{T}L\phi >0\;\mathrm{and}\;∥W∥=1,$$

where *C* is the data covariance matrix and ^*T*^ denotes the transpose. Here, the dependency of *L* on the location is omitted for readability. The equation can be solved analytically [31,72,73], giving the solution

(2)
$$\begin{array}{cc} W & =\frac{C^{-1}L_{\phi }}{\sqrt{L_{\phi }^{T}C^{-2}L_{\phi }}} \end{array}$$


(3)
$$\begin{array}{cc} \phi & =\vartheta_{\min}\left(L^{T}C^{-2}L,L^{T}C^{-1}L\right), \end{array}$$

where 
$\vartheta_{\min}$
 is the eigenvector corresponding to the smallest generalized eigenvalue and 
$L_{\phi }=L\phi$
. Here, we use two methods to estimate the covariance matrix. The first uses the averaged spikes as data [74]. This is called the average-based beamformer in this paper. The second is the event-related beamformer approach, which follows [42] and uses the concatenated single trials as data. The filters are then calculated with Equations (Equation 2) and (Equation 3). The resulting filters are used on the averaged data for localization, independently of the covariance estimation approach. For a point *p* in the source space and the averaged data 
$D_{avg}$
, it is

$$\mathrm{power}\left(p\right)=W_{p,\phi }^{T}D_{avg}D_{avg}^{T}W_{p,\phi }.$$


For regularization, a scaled identity matrix was added to the covariance matrix, as

(4)
$$C_{\mathrm{reg}}=C+\alpha \mathrm{Id}\frac{\mathrm{trace}\left(C\right)}{S}$$

where *S* is the number of sensors. This scaling renders EEG and MEG regularization strength comparable despite the difference in sensor numbers and is standard in all common free toolboxes [75].


$C_{\mathrm{reg}}$
 is then used in place of *C* in the formulas of the beamformer filter (Equation 2) and (Equation 3).

In steps of 
0.002
, we applied regularization from 
α=0
, no regularization, to 
α=0.2
, very strong regularization.

The standard value in the free toolboxes is 
α=0.05
 [75].

#### 2.4.2. Dipole Scan

The dipole scan uses the pseudoinverse 
$L^{+}={\left(L^{T}L\right)}^{-1}L^{T}$
 as a filter [30]. Then, we compute the relative residual variance as

(5)
$$rrv\left(p\right)=\frac{{∥D_{avg}-LL^{+}D_{avg}∥}^{2}}{{∥D_{avg}∥}^{2}}$$


The value for each point is then given as goodness of fit (gof),

(6)
gof(p)=1−rrv(p).


We use no regularization for the EEG and combined EEG/MEG dipole scan. We used the truncated singular value decomposition (tSVD) as regularization method [27,76] for the MEG dipole scan. Thereby, the MEG leadfield was reduced to the two quasi-tangential directions by an SVD. For 
$L=USV^{T}$
, the quasi-radial direction is the singular vector in *V* that corresponds to the weakest singular value. The remaining two singular vectors are used as directions for the new leadfield 
$L_{2}$
, which is then used instead of L.

#### 2.4.3. Comparison of Results

We registered the source space to the postsurgical MRI. We did not observe any problems due to a possible brain shift [77] in the fitting of the source space. Then, we chose all source space points inside the resected resp. thermocoagulated area to represent the resection in our analysis. For simplicity, we will refer to this as “resected” in the remainder of the manuscript. For the beamformer localization, we first analyzed whether the point with maximal power was among those inside the resected area. The distance to the nearest point inside the resection in mm is called the ‘resection distance’. Note that the Euclidean distance between source space points is about 2 mm, so the curves cannot be completely smooth. As a measure of certainty in the localization inside the resection, we divide the maximum power outside the resection by the maximum power inside. This value is called the ‘relative power’. Here, values close to 1 mean uncertainty, while values both larger or smaller than 1 mean more certainty. Values below 1 correspond to a localization inside the resection, with 0 meaning no power outside at all. Values greater than 1 mean an outside result. The function can be arbitrarily large if the power inside the resection is relatively small. However, in praxis, only values around the maximum will influence the evaluation. For the dipole scan, the same principle is used with the goodness of fit instead of the beamformer power estimation.

## 3. Results

### 3.1. Patient 1

The calibrated skull compacta (SC) conductivity was determined as 0.0167 S/m for patient 1. We averaged 248 spikes. The middle of the rising flank was at −6.6 ms. The butterfly plot and topographies at that time point can be seen in Figure 1 and Figure 2, respectively.

#### 3.1.1. EEG

The results for Patient 1 are shown in Figure 3. It should be noted that jumps in the distance measure do not mean large differences in the localization distribution. The beamformer power is not focused on a single spot, but positions with nearly the same power are distributed around it, see Figure 4 or Figure 5, for an example. With changing regularization, these points are weighted differently and the maximum can shift from one to another, leading to a jump in the distance to the resection. There were overall rather low distances for the EEG.

The event-related beamformer behaved similarly for both the 3C and 6C head models, with a distance to the resection volume between 8 mm and 14 mm for every regularization strength and a maximal difference of 5 mm between 6C and 3C. The 6C beamformer’s distance to the resection volume was overall about 2 mm lower and less affected by regularization. The relative power was high for both models, giving them a rather certain localization outside the resection.

The average-based beamformer showed distance values below 5 mm for nearly all regularisation parameters and localized into the resection volume for both 6C and 3C head models when using a regularization parameter of 0.04. To better understand why results do not change smoothly with the regularization parameter, we present in Figure 4 the results for a parameter of 0.04, where the maximum point with maximum power of the localization is inside the resection, but the surrounding points are on the border with nearly the same power. With variations in the regularization, the maximum shifted around these points and gave a localization outside, which explains the changes observed in the distance to the resection volume. As Figure 3 also shows, the localization for the 6C model was inside the resection for all regularization values above 0.142. With a relative power relation value of close to 1, both models showed high uncertainty for regularization parameters above 0.1.

The dipole scans (Dip-3C and Dip-6C in Figure 3 localized into the resection volume for both models with a gof of 97% for each. Their power relation was near 1, showing uncertainty in the localization.

#### 3.1.2. MEG

Except the cases of no regularization and average beamforming in the 6C head model, the differences of the evaluated methods in the distance to the resection volume for the MEG were overall lower than for the EEG. The event-related beamformer in both 6C and 3C head models showed localizations in the resection volume (localization distance of 0 mm) for the main part of evaluated regularization values. The 3C model’s localization distance increased slightly for the higher parameter value choices above 0.128. The 3C model’s power outside the resection was nearly the same as inside, resulting in a power relation close to 1, while the 6C model showed some certainty, with a power relation between 0.96 at low regularization values and 0.92 for high values. Both relations were not strongly affected by regularization. The localization of the 6C model at 0.05 regularization on the inside of the resection border is shown in Figure 5. While the maximum was inside the resection, the signal spread to the middle of the head to some degree.

The average-based beamformer performed worse for both models. Except for very low regularization parameters, the 6C model showed a distance of 6 mm, while the 3C model was at 2 mm. For the regularization parameters from 0.002 to 0.01, the 3C model localized into the resection. No regularization resulted for both models at high distances. The 6C model’s relative power was rather high, giving it a certain localization outside the resection. The 3C model’s relative power switched between a certain localization inside the resection for the low regularization parameters to a very certain localization outside for higher parameters, showing the importance of regularization strength.

The dipole scan in the 6C head model (Dip-6C in Figure 3) localized 4 mm outside the resection, while the 3C scan (Dip-3C) was 2 mm away. Their gof was 88% and 80%, respectively, with very high uncertainty for both as the relative power was close to 1.

#### 3.1.3. MEEG

As shown in the third row of Figure 3, MEEG performed with rather low distances to the resection volume for most methods, but could not reach 0 mm for most parameters.

The 6C event-related beamformer’s distance to the resection volume was 16 mm or 19 mm for the main part of the regularization parameters. Similarly, the 3C model showed a distance above 18 mm for all parameters. Both showed high certainty in their localization.

The 6C averagebased beamformer’s localization distance to the resection volume was with 4 mm distance lowest for regularization parameters between 0.026 and 0.046, while it was considerably higher outside this range. Its relative power showed some variation, but gave certainty to the localization outside the resection for nearly all parameters. The 3C model’s distance to the resection volume showed a large variation for low regularization parameters between 0 and 0.01. For those parameters above 0, the power relation hints at 2 spots with similar power, one inside the resection volume and one 25 mm away from it. The localization stabilized for regularization parameters above 0.036 with a distance to the resection volume of 2 mm.

The dipole scan in the 6C head model (Dip-6C in Figure 3) localized 6 mm outside the resection with gof of 90% and some certainty. The 3C dipole scan (dip-3C) peaked inside with a gof of 80%. Points with high gof closely followed the resection outline, although it crossed the border and thus had a high relative power close to 1. The 3C dipole scan is shown in Figure 6.

### 3.2. Patient 2

The calibrated skull compacta conductivity was determined as 0.0125 S/m for patient 2, i.e., 33% lower than for patient 1. The spike markings for patient 2 showed a slightly different topography between spikes marked in the EEG and MEG. As the MEG spikes showed a less robust and unclear topography with two poles on the border of the sensor cap, the EEG spike markings were used to average not only the EEG, but also the MEG data. Fifty-four spikes were averaged, and the result is shown in Figure 7 and Figure 8. To include the MEG in the analysis, the middle of the rising flank of the averaged spike in the MEG, at −4 ms, was taken as the time point for the analysis. As shown in Figure 7, this meant a later phase of the rising flank of the EEG spike.

The beamformer and dipole scan localization results for patient 2 are shown in Figure 9. Here, the EEG (top row) and MEEG (bottom row) could localize into the resection, while MEG (middle row) could only localize close to the resection, but did not show much activity inside.

#### 3.2.1. EEG

The event-related beamformer with the 6C head model localized into the resection volume with a high certainty for regularization parameters around 0.022. The localization with a parameter of 0.02 is shown in Figure 10. For higher values, both 6C and 3C head models localized 14 mm and 10 mm away, respectively, with strongly increasing certainty.

Without regularization (
α=0
), the average-based beamformers for both head models were 3 mm away from the resection, and an increasing distance to the resection volume can be observed with increasing regularization up to 8 mm and 13 mm for the 3C and 6C head models, respectively. The relative power was well above 1 for all parameters and both head models, thus localizing with certainty outside the resection.

The dipole scans for the 6C and 3C head models localized 29 mm and 31 mm away from the resection with gof of 99% and 98%, respectively. High gof was spread across the whole frontal lobe and was higher in the deeper parts. Both model’s relative power was high.

#### 3.2.2. MEG

For MEG (Figure 9, middle row), the localization distances to the resection volume of the evaluated methods were generally higher than for the EEG, with the best localization still 7 mm away.

For the event-related beamformer, the 3C head model’s distance to the resection volume was 26 mm for the whole parameter range (bf-evt-3C in Figure 9, middle left subfigure). Its relative power was around 2.8 for all parameters, which is outside the shown range (middle right subfigure). The 6C head model (bf-evt-6C) showed an overall lower distance to the resection volume of 20 mm up to a regularization parameter of 0.034 and 18 mm above that value. Its relative power showed slightly more fluctuation, but gave high certainty to a localization outside the resection volume for all parameters. The average-based beamformer in the 6C head model (bf-avg-6C in Figure 9, middle left subfigure) had a distance to the resection volume of 8 mm (regularization parameters between 0.008 and 0.08) and 15 mm (above 0.08), while it was higher at lowest regularization. The 3C head model (bf-avg-3C) performed similarly, with 7 to 8 mm distance to the resection volume for a regularization above 0.026. Lower or no regularization led to very high distance values.

The dipole scans for the 6C and 3C head models localized in 31 mm and 37 mm distance to the resection volume with low gofs of 55% and 54%, respectively. With a relative power of 1.4, both dipoles showed certainty in their localization outside the resection.

#### 3.2.3. MEEG

The results of the combined MEG and EEG source analysis (Figure 9, bottom row) swayed between the EEG and MEG results, improving but also worsening some localizations.

The 6C event-related beamformer’s (bf-evt-6C) results show a similar behavior for MEEG as for the EEG. For low regularization up to a parameter of 0.048, the distance to the resection volume was 3 mm. For higher parameters, the distance jumped to 24 mm, with a fast increase in certainty. The 3C model showed similar results, although it could localize inside the resection for a range of parameters and the increase in its distance only happened at a higher regularization parameter of 0.09.

The MEEG average-based beamformer for both head models (bf-avg-6C and bf-avg-3C) showed a comparable performance as for single modality EEG and MEG. For both head models, distances to the resection volume decreased with increasing regularization strength and stabilized at a distance of 8 mm for regularization parameters above 0.03. Their relative power was almost identical and rather high.

In Figure 11, the MEEG average-based beamformer localization at 0.05 regularization strength is presented.

The performance of the MEEG dipole scans for the 3C head model (Figure 9, bottom row, dip-3C) followed the EEG with a distance of 31 mm and a gof of 79%. The 6C head model’s localization was improved in comparison to the single modalities, although its distance was still 21 mm with a gof of also 79%. For both head models, the dipole scan showed certainty in the localization outside the resection volume.

## 4. Discussion

In a recent review [78], EEG and MEG source analysis were found to have a relatively high sensitivity but low specificity for the identification of the epileptogenic zone, so that studies allowing unbiased evaluation to determine the added value and diagnostic accuracy of source analysis in the presurgical workup of refractory focal epilepsy are needed. In this work, we investigated the influence of the choice of forward and inverse methods in EEG, MEG and combined MEEG source analysis [15] in two case studies with focal epilepsy patients. The patients were both suffering from refractory focal epilepsy at the time point of the EEG/MEG measurement of interictal epileptiform activity, were diagnosed with focal cortical dysplasia (FCD) type IIb and had successful surgery with seizure-free outcome more than three years post-surgery. Based on [35,79], our hypothesis was that, due to successful surgeries, the resection volumes, or more specifically, the distance of source localizations to the lesions according to MRI and the resection volumes, could be used as a validation criterion of the different investigated source analysis approaches.

On the forward modelling side, we compared a standard three compartment (3C) [15,30,54] with a calibrated six compartment (6C) anisotropic head modelling approach [6,47,54]. The calibration was done for individual skull conductivity using the somatosensory P20/N20 component, since skull conductivity is known to strongly vary inter- and intra-individually [6,22,64,65,66,67] and was found to be the most sensitive conductivity parameter on the EEG side [25,47]. For the inverse problem, we evaluated the performance of dipole (or deviation) scanning [30] and beamforming [31] with different covariance matrix estimation approaches and at different levels of regularization strength [72,73]. For beamforming, we compared an *event-related beamformer* approach [42] with an *average based beamformer*, i.e., a unit-noise-gain beamformer [72,73,80] applied on the averaged data, and we used the maximum of the beamformer power for localization.

### 4.1. Difference between the Patients

The factors with potential influence on source localization that we investigated were the modality, volume conduction model, the beamformer method, and corresponding regularization strength. However, these differences have to be understood in relation to the differences between the patients and their data. While both patients suffered from an FCD, their location was very different. Patient 1’s FCD was located in the right superior parietal lobule. The topography of the averaged spikes was dipolar in both EEG and MEG with a maximum goodness-of-fit (gof) of the dipole scan of 97% and 88%, and both localized into the resection or very close to it, respectively (see Figure 3, left column). Due to an oblique source orientation, both EEG and MEG are sensitive to the underlying source activity [13,14,15]. In contrast, Patient 2’s FCD was a small bottom-of-sulcus FCD in the anterior part of the left superior frontal sulcus. The EEG topography points to a lateral left frontal and strongly radially oriented source (dipole scan gof of up to 99%). While the MEG signal is always weak for radial sources [13,14,15], in this case, it was combined with a missing sensor coverage at the front of the head (Figure 8, right), resulting in a MEG dipole scan gof of only max 55%. Due to the smaller size of the FCD and the resulting thermocoagulated area, the distances for patient 2 can additionally be expected to be larger. On the other side, this means that low values point to a localization very close to the FCD, which may thus be an especially good methodological validation and clinically specifically valuable. With these premises, the two cases presented very different challenges to both the modalities and methods.

Still, all methods showed good results and high concordance in patient 1 and nearly all did so in patient 2.

### 4.2. Beamformer Parameters and Localization

In contrast to the dipole scan, beamformers can localize ongoing activity and are not dependent on time-locked data. However, they profit from high SNR and thus averaging, when appropriate. For epileptic spikes, clustering of single spike localization might show valuable information about the extent of the irritative zone [71,81,82], but averaging of similar spikes increases the SNR and improves the reliability of the reconstruction [71,83,84]. In our study, we try to localize the center of gravity of the spike activity, which is easier to compare to the standard dipole scan approach. Both beamformer methods we investigated try to optimize the beamformer design to event data, but differ in the estimation of the covariance matrix. The average-based beamformer is a straight application of the beamformer approach [85] on the averaged data and has been shown by [74] to give similar results to dipole scans on fields evoked by median nerve stimulation. By using the concatenated single trials instead of the averaged data for the estimation of the covariance matrix, the event-related beamformer [41,42] approach offers a more flexible approach. It can be used with events that are time-locked and those, that evoke a synchronization. It was first introduced by [41] for motor tasks and has been used successfully for epilepsy data [86]. Increasing the time sample size should improve the covariance estimation [40] and decrease the need for regularization [40,41,42]. However, the SNR of the single trials will be lower, and the covariance matrix might include noise components that would have been removed by averaging first [41,42]. In our investigation, neither approach showed a clear advantage. While we see a tendency for the average-based approach to be more stable across modalities, it did not necessarily localize closer to the resection (see left column in Figure 3 and Figure 9). For both MEG and MEEG, using no regularization resulted in high distances for the average-based approach, while the event-related approach had much smaller variations at low regularization strength, which seems to confirm the ideas of [41,42]. For higher strength, both methods showed a similar behavior, with usually low changes in the distance to the resection, although some jumps of 1 cm to 2 cm have appeared, mostly in the MEEG. This seems to indicate that the scaling [75], shown in Equation (Equation 4), works as intended. In the two cases, the optimal regularization strength was lower than the quasi standard of 0.05 [75], which is used by all open source toolboxes. For all modalities, parameter values in the range of 0.02 to 0.04 performed better, even though 0.05 would have still given good results (Figure 3 and Figure 9). Overall, we can only recommend applying some regularization, especially with the average-based approach, and to report on the values used.

Except for the estimation of the covariance, we used the same filter design for both approaches. As the linear constraint of the filter introduces a depth bias, the depth normalization by the filter norm or a noise covariance matrix was introduced, called ‘neural activity index’ [85] or ‘Pseudo-Z value’ [87]. The unit-noise gain filter that we used gives an analytical formulation of a filter with the same output, but in a single step [72]. This enables the derivation of an optimal source direction, which increases output SNR and is fast to compute [72].

In comparison to the dipole scan, the beamformer showed comparable accuracy for patient 1 and better results for patient 2. Our findings in the examined two patients are concordant with [86], showing that beamformers are a reliable way to localize epileptic spikes.

### 4.3. Models and Modalities

We compared the performance of a three compartment and a six compartment model with the same source space. In most cases, neither model had a significant advantage to the other and better performance in one case was offset with worse performance in another. Often, the models showed a shifted behavior, with the same tendencies at higher or lower regularization.

From single dipolar source simulations, differences from a few millimeters to centimeters have to be expected between 3C and 6C models [17,80], which fits to our results with differences from millimeter range to maximally 2.5 cm (event-related MEEG beamformer in Figure 9). Ref. [88] found accurate skull modelling to be necessary for EEG source modelling, with especially high errors near suture line. Since we investigated only the distance to the resection and not between the foci of the methods, these distances cannot be precisely inferred. For example, the EEG dipole scan for patient 1 localized into the resection volume for both models, but did not localize the same position. Furthermore, we only evaluated localization and not the source orientation and strength differences, which are known to be considerably influenced by volume conduction differences between isotropic 3C and anisotropic 6C models [6,17,24,25]. Source orientation, for example, might also be important in addition to absolute localization in attributing epileptic activity to subcompartments of the respective brain area [49,89,90].

In both patients, the calibrated skull conductivity in the 6 compartment model, i.e., 0.0167 S/m for patient 1 and 0.0125 S/m for patient 2, was relatively close to the standard value of 0.01 S/m used in the 3 compartment model. This helps to explain why the localization differences between the 3C and 6C models were often only in the millimeter range, when knowing that skull conductivity is the most sensitive conductivity parameter for the EEG source localization [25,64,65]. It should be noted here that this relation is specific to these patients and should not be generalized, as calibrated skull compacta conductivity values of 0.0033 S/m [91] and 0.0024 S/m [54] and an age-dependency with, however, a large interindividual variability [6], have been found.

Our source space was limited to the gray matter surface for both 6C and 3C head models, which is also useful to improve the realism of the reconstruction, since EEG and MEG dipolar sources can only be found in the gray matter compartment [15]. For the orientation of the sources, we used no constraints. While the sources should be perpendicular to the cortex surface, at least in a healthy brain [15,45,92,93], the constraint inhibits the localization quality when the modelling is not exact. In [94], the beamformer quality deteriorated strongly with errors in MRI registration and the estimation of the cortex surface. Additionally, errors in the head modelling, especially conductivity modelling, will further influence the localization [80]. This is why we do not recommend a normal constraint, even if on the other side, a dimension reduction could have a positive effect on the SNR of the beamformer reconstruction [72,94].

Both MEG and EEG gave good results for our patients, even if, with the investigated parameters, the MEG performance was slightly more robust to regularization. The combined analysis of MEG and EEG gave comparable and satisfying results with respect to the clinical outcome, further strengthening the contribution of non-invasive source analysis to presurgical epilepsy diagnosis and to guiding the positioning of sEEG or ECoG grids. With our imprecise error measure (see below) and the good results for the single modalities, a strong improvement like seen in [71,91] was unlikely. In addition, Ref. [83] reported a strong improvement of MEG localization results of anterior temporal spikes when a few EEG electrodes were included. Our results thus further motivate combined MEEG source analysis, while we also showed that, due to the remaining model inaccuracies, combined MEEG results should always be compared to results of single modality EEG or MEG source analysis.

### 4.4. Limitations of Our Error Criterion

Finally, it is surely very important to understand the limitations of our source analysis validation criterion, i.e., the distance to the resection volume. In the case of FCD, the irritative zone might only be neighboring the FCD and therefore the resection volume. A reason for this might be that the pyramidal cells inside the FCD volume might not build an open field structure, which is, however, necessary to produce activity that can be marked in EEG and MEG [45,92,93]. In such a case, a source analysis result with a considerable distance to the resection volume might still be accurately reconstructing the irritative zone. Furthermore, the MRI correlate of an FCD IIb, which significantly influences the definition of the resection volume, predominantly corresponds to the distribution of pathologic balloon cells [45,95,96], which however are electrophysiologically silent [45,97]. In contrast, dysmorphic neurons, the putative epileptogenic cell population, show an overlapping, but importantly not identical distribution [45,96]. Their MRI correlate is, however, subtle: inhomogeneities of the intracortical signal [96,98] and blurring of the gray-white matter junction due to dyslamination [99]. Correspondingly, these discrepancies introduce further potential variability between the location of the epileptogenic zone, the MRI finding and correspondingly the resection volume. All results in this work thus have to be seen in the light of these limitations to our validation criterion, while on the other side there might be no better one than the use of the resection volumes in seizure-free patients.

## 5. Conclusions

Beamformers can be used to localize epileptic spikes. In some cases, they can outperform dipole scans, but the real activity is hard to distinguish when the localization of different methods differ. Using the covariance of the average data gave more stable results than using concatenated trials. The difference between our 3 and 6 compartment models were only in the millimeter to centimeter range, most probably due to very similar skull conductivity and the usage of a gray matter source space without orientation constraint. Regularization strength is an important factor, especially when only the maximum position of the beamformer localization is used. While not optimal in the presented cases, the standard value of 0.05 gave good results. 

## Figures and Tables

**Figure 1 brainsci-12-00114-f001:**
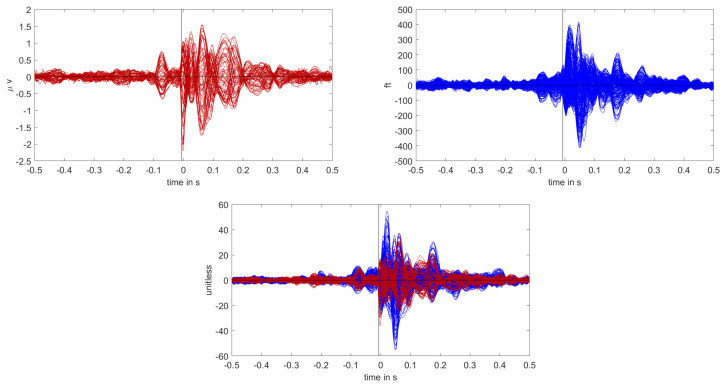
Butterfly plot of the averaged spikes of patient 1 in the EEG (red), MEG (blue), and MEEG (blue and red). The time point of the rising flank at −6.6 ms is marked with a vertical line from above and below. The horizontal line marks the zero activity line.

**Figure 2 brainsci-12-00114-f002:**
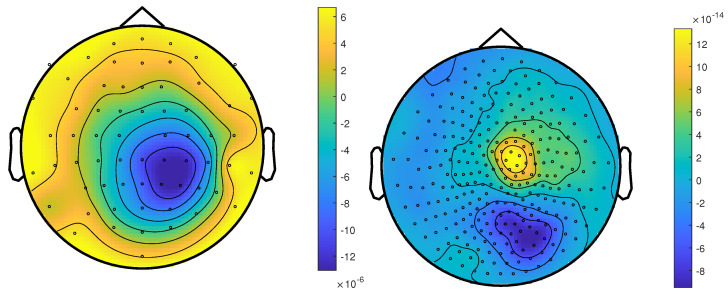
The topographic maps of patient 1, left EEG and right MEG, show the potential and field amplitudes with yellow for positive and blue for negative at the sensor level relative to the mean at the time point of the middle of the rising flank at −6.6 ms. The sensors are interpolated to a sphere model that represents the head.

**Figure 3 brainsci-12-00114-f003:**
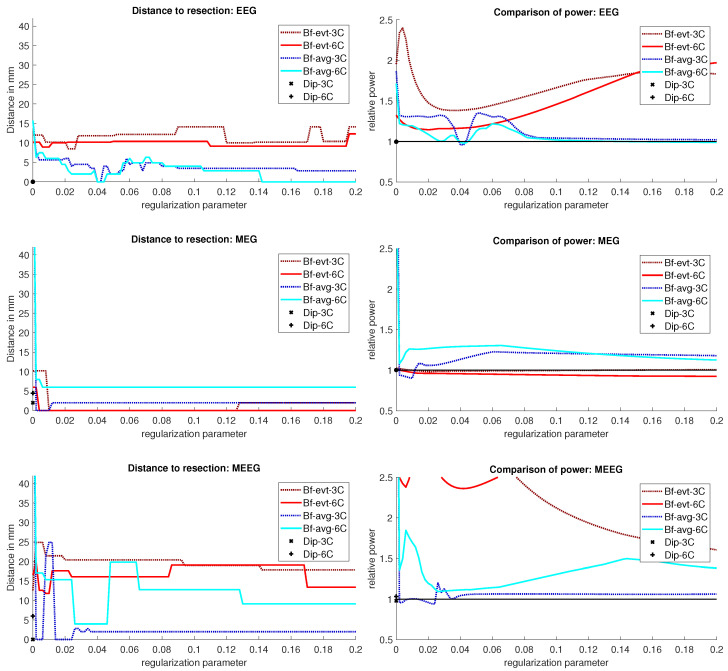
Localization results for patient 1. The distance of the localization of the beamformer maximum to the resection volume for each regularization value from 0 to 0.2 in steps of 0.002 is shown on the left. The corresponding power relation, which is the maximum power outside the resection volume divided by the maximum inside, is depicted on the right. Each pair of graphs is specific to one modality: EEG, MEG, and MEEG, from top to bottom. The methods are color coded in the same way for all modalities: lighter colors and full lines are used for the 6 compartment, darker dotted lines for the 3 compartment head model.

**Figure 4 brainsci-12-00114-f004:**
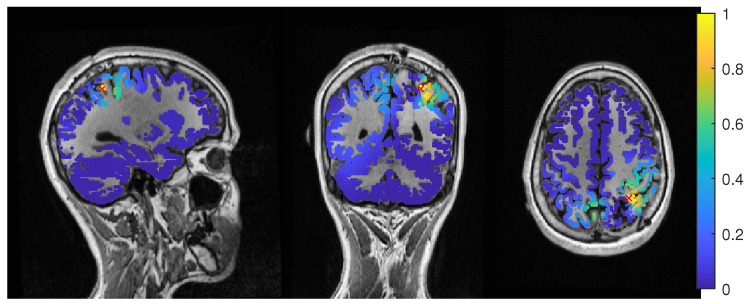
Patient 1’s result of the EEG average-based beamformer in the 6 compartment head model at 0.04 regularization with sagittal (**left** subfigure), coronal (**middle** subfigure) and axial (**right** subfigure) cuts through T1 MRI and source space. Note that the MRI is not in radiological convention.

**Figure 5 brainsci-12-00114-f005:**
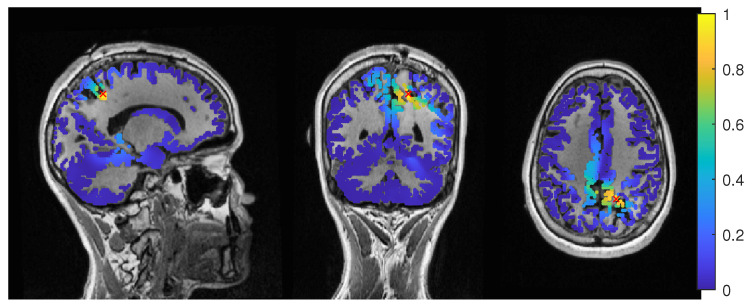
Patient 1’s result of the MEG event-related beamformer in the 6 compartment head model at 0.05 regularization with sagittal (**left** subfigure), coronal (**middle** subfigure), and axial (**right** subfigure) cuts through T1 MRI and source space. The maximum is additionally marked with a red x. Note that the MRI is not in radiological convention.

**Figure 6 brainsci-12-00114-f006:**
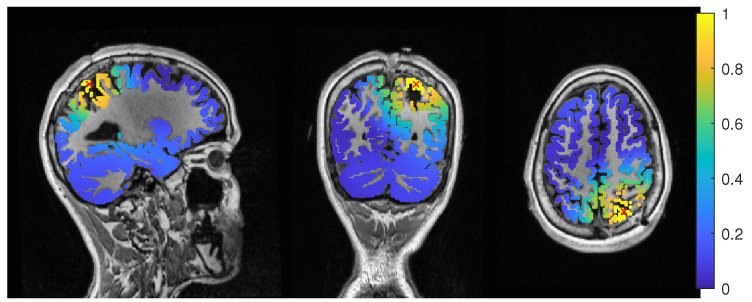
Patient 1’s gof of the MEEG dipole scan in the three compartment head model with sagittal (**left** subfigure), coronal (**middle** subfigure) and axial (**right** subfigure) cuts through T1 MRI and source space. The maximum is additionally marked with a red x. Note that the MRI is not in radiological convention.

**Figure 7 brainsci-12-00114-f007:**
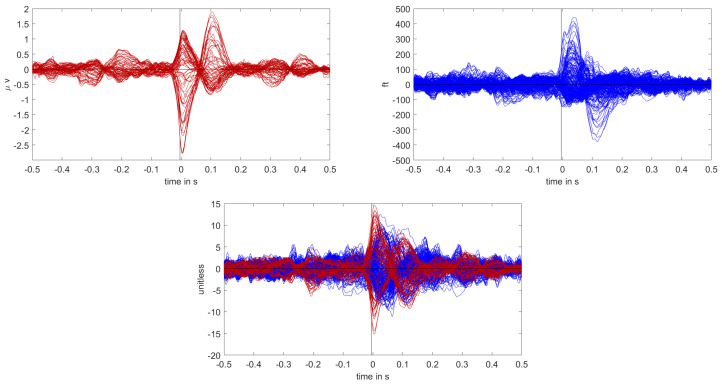
Butterfly plot of the averaged spikes of patient 1 in the EEG (red), MEG (blue), and MEEG (blue and red). The time point of the rising flank at −4 ms is marked with a vertical line from above and below. The horizontal line marks the zero activity line.

**Figure 8 brainsci-12-00114-f008:**
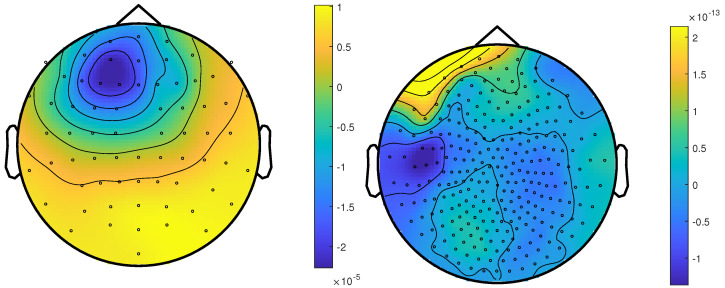
The topographic maps of patient 2, left EEG and right MEG, show the potential and field amplitudes with yellow for positive and blue for negative the sensor level relative to the mean at the time point of the middle of the rising flank at −4 ms. The sensors are interpolated to a sphere model that represents the head.

**Figure 9 brainsci-12-00114-f009:**
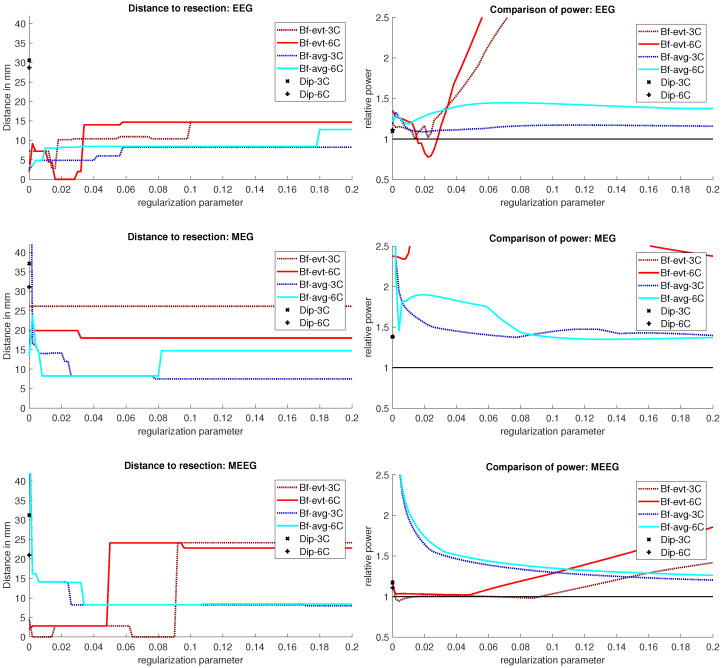
Localization results for patient 2. The distance of the localization of the beamformer maximum to the resection volume for each regularization value from 0 to 0.2 in steps of 0.002 is shown on the left. The corresponding power relation, which is the maximum power outside the resection volume divided by the maximum inside, is depicted on the right. Each pair of graphs is specific to one modality: EEG, MEG, and MEEG, from top to bottom. The methods are color coded in the same way for all modalities: lighter colors and full lines are used for the 6 compartment, darker dotted lines for the 3 compartment model.

**Figure 10 brainsci-12-00114-f010:**
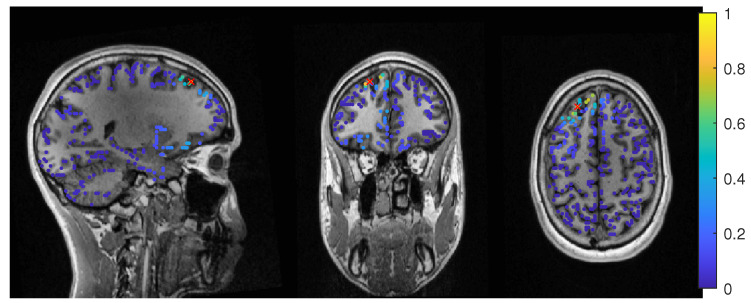
Patient 2’s result of the EEG 6 compartment event-related beamformer at 0.02 regularization with sagittal (**left** subfigure), coronal (**middle** subfigure) and axial (**right** subfigure) cuts through T1 MRI and source space. The maximum is additionally marked with a red x. Note that the MRI is not in radiological convention.

**Figure 11 brainsci-12-00114-f011:**
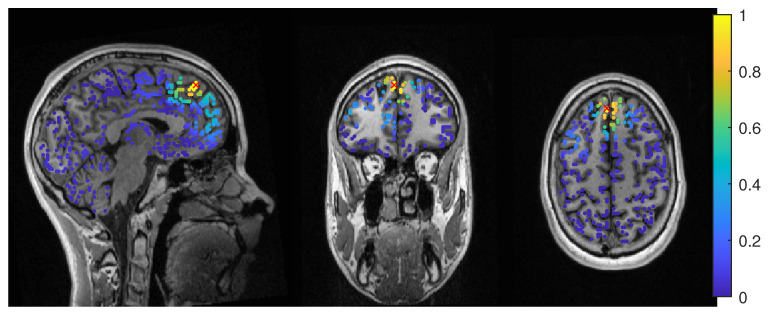
Patient 2’s result of the MEEG average-based beamformer in the 6 compartment head model at 0.05 regularization with sagittal (**left** subfigure), coronal (**middle** subfigure) and axial (**right** subfigure) cuts through T1 MRI and source space. The maximum is additionally marked with a red x. Note that the MRI is not in radiological convention.

## Data Availability

The data are not publicly available due to patient privacy. Codes are available on request from the corresponding author.

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
