# Peer review of "Validating EEG, MEG and Combined MEG and EEG Beamforming for an Estimation of the Epileptogenic Zone in Focal Cortical Dysplasia"

_brainsci, 2022, doi:10.3390/brainsci12010114_

Round 1

Reviewer 1 Report

This is the study about new methodology using both EEG and MEG beamforming. This combination method can support the weak points of EEG or MEG. The weak points of this study is a few cases of the study, however, the methodology and discussion are well established. I think that this article did not have the points to be improved. I think that it can be published on the journal.

Author Response

We edited the grammar and style of the paper. Changes are marked in the new manuscript with the tag “English”.

We added one further citation for the importance of head modelling in the discussion, and clarified some minor points. Otherwise, the manuscript is the same.

Reviewer 2 Report

This validation study compares the standard dipole scanning method with two beamformer approaches, and it analyzes the influence of the covariance estimation and the strength of regularization on the localization performance for EEG, MEG and combined EEG and MEG. 

The article is well written and some new findings are obtained, which enriches the research on MEG and EEG source analysis.

My comments are the following:
1. The study uses only two patients, which may limit the generality. More patients may be helpful for the study. The authors should clarify why only two patients are used.
2. The labels of the x-axis in Figure 3 and Figure 9 should be added. 
3. Please carefully check the text to remove some issues of language or typos. For example, 
-L123, "analysis a a very small "
-L333,  "The dipole scan (Dip-3C and Dip-6C in figure ..."
...

Author Response

We edited the grammar and style of the paper. Changes are marked in the new manuscript with the tag “English”. We added the missing labels in the figures.

We added one further citation for the importance of head modelling in the discussion, and clarified some minor points. Otherwise, the manuscript is the same.

We agree that the small number of patients is a limitation, but we had to decide between an in depth analysis with few patients or a more generalized analysis with more, and decided for the few.

Here, the requirements for the patients were:

Epilepsy with an FCD type IIb,

combined MEG and EEG measurements, which is non-standard in the diagnosis,

enough spikes during the MEG/EEG measurement,

surgery with successful outcome at least one year later,

MRI measurement before and after surgery,

patient's consent to the study.

Each of these points limited the available patient data, but was also needed for the analysis. We are working to acquire more patients, and we will work to include more in future studies.

This manuscript is a resubmission of an earlier submission. The following is a list of the peer review reports and author responses from that submission.